# Competitive Exclusion Bacterial Culture Derived from the Gut Microbiome of Nile Tilapia (*Oreochromis niloticus*) as a Resource to Efficiently Recover Probiotic Strains: Taxonomic, Genomic, and Functional Proof of Concept

**DOI:** 10.3390/microorganisms10071376

**Published:** 2022-07-08

**Authors:** Javier Fernando Melo-Bolívar, Ruth Yolanda Ruiz Pardo, Howard Junca, Hanna Evelina Sidjabat, Juan Andrés Cano-Lozano, Luisa Marcela Villamil Díaz

**Affiliations:** 1Doctorado en Biociencias, Faculty of Engineering, Universidad de La Sabana, Campus Universitario del Puente del Común, Km 7 Autopista Norte de Bogotá, Chía 250001, Colombia; javiermebo@unisabana.edu.co (J.F.M.-B.); ruth.ruiz@unisabana.edu.co (R.Y.R.P.); juancanloz@unisabana.edu.co (J.A.C.-L.); 2RG Microbial Ecology: Metabolism, Genomics & Evolution, Div. Ecogenomics & Holobionts, Microbiomas Foundation, Chía 250001, Colombia; howard.junca@gmail.com; 3Menzies Health Institute Queensland, Griffith University, Gold Coast, QLD 4222, Australia; h.sidjabat@griffith.edu.au

**Keywords:** continuos-flow competitive exclusion culture (CFCEC), probiotics, microbiome, freshwater fishes, whole-genome sequencing, Nile tilapia, *Lactococcus lactis*, *Bacillus*, *Streptococcus agalactiae*, *Aeromonas hydrophila*

## Abstract

This study aims to mine a previously developed continuous-flow competitive exclusion culture (CFCEC) originating from the Tilapia gut microbiome as a rational and efficient autochthonous probiotic strain recovery source. Three isolated strains were tested on their adaptability to host gastrointestinal conditions, their antibacterial activities against aquaculture bacterial pathogens, and their antibiotic susceptibility patterns. Their genomes were fully sequenced, assembled, annotated, and relevant functions inferred, such as those related to pinpointed probiotic activities and phylogenomic comparative analyses to the closer reported strains/species relatives. The strains are possible candidates of novel genus/species taxa inside *Lactococcus* spp. and *Priestia* spp. (previously known as *Bacillus* spp.) These results were consistent with reports on strains inside these phyla exhibiting probiotic features, and the strains we found are expanding their known diversity. Furthermore, their pangenomes showed that these bacteria have indeed a set of so far uncharacterized genes that may play a role in the antagonism to competing strains or specific symbiotic adaptations to the fish host. In conclusion, CFCEC proved to effectively allow the enrichment and further pure culture isolation of strains with probiotic potential.

## 1. Introduction

The need to include high-quality proteins increases as the world’s population grows and fisheries and aquaculture have become excellent sources. More than 61.04 million people worldwide are directly involved in the fishing industry, with fishing accounting for 63.4% and aquaculture accounting for 36.6% [1]. Fish fillet or meat is a nutritious alternative since it contains essential amino acids, unsaturated fat, and minerals [2,3]. Increased cultured fish production requires the advancement of technology and management strategies in aquaculture [4]. However, large-scale production in aquaculture facilities exposes fish to stressful environments, making them more susceptible to disease outbreaks and significant financial losses [5,6]. Antibiotics are traditionally recognized to control diseases and increase body weight by enhancing the feed conversion efficiency, which directly increases aquaculture’s overall output [7,8]. Nevertheless, antibiotic misuse has harmful effects on the environment and fish health, can induce antibiotics resistance to antibiotics on fish pathogens and surrounding ecosystems bacteria, including zoonotic pathogens [2,9,10,11,12,13].

Numerous biotechnological approaches have been evaluated to decrease the detrimental effects of antibiotics on animals and the environment, and it is consequently on to the consumers. Non-specific immunostimulants, vaccines, probiotics, prebiotics, symbiotics, medicinal plants, and other ways to regulate fish diseases may be included in these aquaculture biotechnologies [14,15]. Probiotics have a unique potential among these biotechnologies since they combine the benefits of many approaches and are considered environmentally friendly [16]. Probiotics as defined as “any microbial cell provided via the diet or rearing water that benefits the host fish, fish farmer or fish consumer, which is achieved, in part at least, by improving the microbial balance of the fish composition which maintains the homeostatic of the fish” [17]. Probiotics can promote growth and disease resistance, improve water quality, and host nutrition through digestive enzyme production, competitively eliminate harmful bacteria, and improve survival rate and stress tolerance [18,19,20].

When selecting potential probiotic strains, various criteria to allow the survival of the strains in the gastrointestinal tracts should be taken into consideration, including acid and bile resistance, gastric juice survival, extracellular enzyme production, production of antimicrobial substances that inhibit pathogen growth in vitro, the ability to adhere to intestinal mucus and colonize the gut as well as fulfill biosafety requirements (hemolytic activity, antibiotic susceptibility, and others) [21].

It has recently been suggested that microbial safety assessment includes identifying the selected strains, ideally with whole-genome sequencing and bioinformatic analysis to screen potential antibiotic-resistant genes, toxin-coding genes, and other potentially harmful features [19]. Also, a pivotal aspect in this process is the actual biological resource harboring the probiotic strains and the ways used to enrich and isolate such bacteria in pure cultures. 

Even though phenotypic identification of probiotic bacteria using standard microbiological methods is important for identifying species, it is not always reliable. Also, these methods alone cannot tell the difference between many species [22]. Lastly, species-level identification needs to be done with reliable molecular techniques [23]. Therefore, studying bacterial whole genome sequencing is essential to bacterial identification and functional annotation of bacteria with probiotic activity [24]. 

In a previous study, a competitive exclusion of bacterial culture using autochthonous tilapia gut microbiome samples as an initial source was enriched and developed in lab bioreactors giving rise to a strong mixed antibacterial activity against *Streptococcus agalactiae* [25]. In this work, we are reporting selected bacterial strains that are derived from such Nile tilapia gut microbiome continuous-flow competitive exclusion culture (CFCEC) as a proof of concept showing the feasibility of this particular resource and rational approach to efficiently select and expand the set of probiotic isolates from and for tilapia, and their particularities from taxonomic and functional standpoints. Furthermore, these strains are characterized in greater detail regarding advantageous probiotic traits, such as survival in gastrointestinal conditions, genome sequencing, annotation and functions, and comparative phylogenomics to the genomes of publicly available isolates of the same species.

## 2. Materials and Methods

### 2.1. Ethical Statement

The project followed the Colombian national government’s regulations. The Permit for accessing genetic resources was issued by the Colombian Ministry of Environment Number 117 (Otrosí 4) on the 8th of May 2018 for five years.

### 2.2. Bacteria Selection

We previously described and analyzed CFCEC by culture-dependent and independent culture means [25]. Three probiotic bacteria A12, M4, and M10, were isolated from a competitive exclusion culture derived from juvenile tilapia intestinal content previously. These bacteria were selected based on the potential antimicrobial activity against pathogenic fish bacteria. These bacteria were activated by inoculating them onto Tryptose Soy Agar (TSA, Scharlab S. L., Barcelona, Spain) and incubated at aerobic atmosphere for 24 h at 28 °C in preparation for further use.

### 2.3. Whole-Genome Sequencing

#### 2.3.1. DNA Extraction, Library Preparation, and Sequencing

Bacterial isolates A12 and M4 were submitted to the Forensic and Scientific Services, Queensland Health, Australia, for DNA extraction, library preparation, and whole-genome sequencing. Bacterial genomic DNA was extracted using the DSP DNA Mini Kit (Qiagen, Hilden, Germany). The Nextera XT DNA Library Prep Kit (Illumina, Melbourne, VIC, Australia) was used to prepare the sequencing library, which then was followed by sequencing on the NextSeq 500 (Illumina) using the NextSeq 500 Mid Output v2 kit (Illumina).

DNA of bacterial M10 was extracted using the DNeasy^®^ UltraClean^®^ Microbial Kit (Qiagen, Hilden, Germany) following the manufacturer’s instructions. Briefly, bacteria from frozen stock were grown on TSA at 28 °C for 24 h. This DNA extraction method was optimized to reach a DNA concentration of 100 ng/uL with the 260/230 ratio as an indicator of DNA purity > 1.8. Firstly, colonies were harvested using 10 uL disposable loops for approximately two loopful, suspended in the Powerbead solution, and transferred into a Powerbead tube. The protein precipitation step was followed by the DNA binding, washing, and elution steps. Finally, the DNA quantification was done with a Nanodrop 1000 (Thermo Fisher Scientific, Scoresby, VIC, Australia). The DNA of bacteria isolated M10 was sent to Macrogen (Seoul, Korea) for library preparation and whole-genome sequencing. The sequencing library was prepared using the TruSeq Nano DNA kit. Finally, the paired reads were sequenced using the Illumina platform NovaSeq 6000.

#### 2.3.2. Quality Control, Trimming, Assembly of Paired Ends Reads, and Contigs Selection

Shovill v1.1.0 (SPAdes, v3.15.3; Velvet, v1.2.10; Megahit, v1.2.9; Skesa, v2.4.0; using default arguments; https://github.com/tseemann/shovill (accessed on 2 January 2022)) [26] was used to perform quality control (QC), filtering, trimming, and de novo assembly on raw WGS data (FASTQ files). After obtaining the assembled contigs, a Quast comparison was used to select the assembly with the fewest contigs and an N50 length of near 50% of the overall genome length.

#### 2.3.3. Whole-Genome Sequencing Identification

JSpeciesWS v3.8.5 was used to identify the species (default parameters) by Tetra correlation search in conjunction with ANIb (average nucleotide identity, calculated using the BLAST algorithm) and ANIm (average nucleotide identity, computed using the MUMmer alignment tool) [27]. In addition to this approach, the complete genome drafts were analyzed using the Type (Strain) Genome Server (TYGS) [28] to define taxonomical affiliation at the species level.

#### 2.3.4. Order, Orientation, and Scaffolding of Contigs Assembled

Following identification, RefSeq (NCBI Reference Sequence Database) was used to obtain each bacteria’s reference genomes (A12, M4, and M10). Then, the assembled contigs were uploaded with their respective reference genomes to Medusa (http://combo.dbe.unifi.it/medusa (accessed on 8 January 2022)) [26] to determine orientation and order among contigs to produce longer scaffolds [29,30]. The genome sequence data were uploaded to the Type (Strain) Genome Server (TYGS), a free bioinformatics platform available under https://tygs.dsmz.de (accessed on 19 May 2022), for a whole genome-based taxonomic analysis [28]. The analysis also used recently introduced methodological updates and features [31]. Information on nomenclature, synonymy, and associated taxonomic literature was provided by TYGS’s sister database, the List of Prokaryotic names with Standing in Nomenclature (LPSN, available at https://lpsn.dsmz.de (accessed on 19 May 2022)) [31]. The TYGS provided the results on 13 May 2022. For determination of closely related type strains, pairwise comparison of genome sequences, phylogenetic inferences, and type-based species and subspecies clustering.

#### 2.3.5. Genome Comparison

Sequences from the microorganisms *Lactococcus lactis* subsp *lactis*, *Priestia megaterium*, and *Priestia aryabhattai* were downloaded from the RefSeq database (https://www.ncbi.nlm.nih.gov/refseq/ (accessed on 9 January 2022)) [32]. We compared the number of CDS, genome length, number of rRNAs, tmRNA, and tRNA in each bacteria using a database we generated using reference genomes (from *Lactococcus* for A12 or *Priestia* for M4 and M10) annotated in Prokka [33]. The Roary [34] program was run in Galaxy (https://usegalaxy.org/ (accessed on 17 January 2022)) with Prokka’s gff files to compare the genome’s source to determine the core and cloud genomes. The core genome alignment file for Roary was used to generate a phylogenetic tree using the free program IQtree. Finally, the most closely related genomes were compared using an upset plot produced in R with a matrix derived from Roary’s presence-absence genes.

#### 2.3.6. Functional Annotation

To find genes involved in amino acid biosynthesis, vitamin biosynthesis, carbohydrate metabolism, adhesion, and aggregation, among other features, Uniprot database (https://www.uniprot.org/ (accessed on 19 January 2022)) [35], and Bagel4 [36] (bagel4.molgenrug.nl/index.php (accessed on 19 January 2022)) were used to identify bacteriocins. The identified genes were then aligned directly into the genomes of A12, M4, and M10 using the RAST [37,38] genome comparison tool to confirm their presence in the genomes. Finally, A Blastx (https://blast.ncbi.nlm.nih.gov/Blast.cgi (accessed on 2 February 2022)) [39] was used to reannotate the gene identified in our genomes.

#### 2.3.7. Data Availability

The raw reads used to assemble the draft genome were deposited in the Sequence Read Archive (SRA) under PRJNA577063. The genome sequence data of M10, A12 and M4 are deposited with Accession numbers of SRX15182551, SRX6979445, SRX6979446, respectively.

### 2.4. Probiotic In Vitro Characterization

#### 2.4.1. Antibacterial Activity against Streptococcus Agalactiae and Aeromonas Hydrophila

The antibacterial activity of the bacteria selected was determined according to the methodology of Villamil, et al. [40]. Briefly, the probiotic bacteria were incubated in TSB or MRS overnight at 28 °C. Then, the bacterial culture was adjusted to pH 3.5 with 2 N HCl. After that, the culture was heated at 80 °C and centrifuged at 500× *g* for 30 min. Followed by adjusting the extracellular products (ECPs) to pH 6.5 and the 0.22 µm filter was used to sterilize. Finally, the pathogenic bacteria *Streptococcus agalactiae* and *Aeromonas hydrophila* were used to evaluate the antibacterial activity of the probiotic bacteria in a 96 wells plate by dispensing 50 µL of the bacterial suspension (10^8^ cells/mL) and 50 µL of the ECPs. After 18 h of incubation, changes in optical density (600 nm) were measured. 

#### 2.4.2. Antibiotic Minimal Inhibitory Concentration Determined by Etest Method

The minimal inhibitory concentration of bacteria against selection of antibiotics (Streptomycin, Ciprofloxacin, Kanamycin, Tetracycline, Ampicilin, Vancomycin, Clindamycin, Gentamicin, Cloramphenicol) was evaluated according to Florez, et al. [41]. Briefly, individual colonies were suspended in sterile glass or plastic tubes containing 2–5 mL of sterile saline until a density corresponding to McFarland standard 1 or its spectrophotometric equivalent (approximately 3 × 10^8^ CFU/mL) was obtained. A sterile cotton swab of the above McFarland suspension was spread on Mueller Hinton agar plates. After approximately 15 min for the agar surface to dry, the Etest strips were applied (BioMérieux, Durham, NC, USA). After 48 h of incubation at 28 °C for A12, M4, and M10. The results were classified as Resistant (R) or susceptible (S) using the EFSA-recommended cut-off value according to the respective species [42].

#### 2.4.3. Hemolytic Activity

Blood agar plates and hemolysis protocols proposed by Buxton [43] were used to perform the hemolytic activity. The blood agar was prepared using Trypticase Soy Agar (TSA) culture medium as the base medium (Scharlau, Barcelona, Spain). After preparing and sterilizing the base medium (TSA), it was brought to a temperature of 45–50 °C, and sterile defibrinated sheep blood was added at a concentration of 5% (*v*/*v*) and mixed thoroughly. The solution was shaken vigorously to incorporate all components, and the medium was poured into Petri dishes. The bacteria were seeded onto the agar after the culture medium was solidified. The petri dishes were incubated at 28 °C for 48 h. A clear zone shows β hemolysis around the colonies, α hemolysis is shown by a greenish zone around the colonies, and no clear zone shows γ hemolysis around the colonies, which is a safe non-hemolytic isolate [44].

#### 2.4.4. Bile Salts and pH Survival

With some changes, this test was conducted following Muthukumar and Kandeepan [45], and Aragón-Rojas, et al. [46]. First, the Brain Heart Infusion Broth (BHI, Scharlau, Barcelona, Spain) culture medium was prepared for pH resistance, by adding 1 N HCl to a final pH of 2.0 or 3.0. 

Bacterial survival in bile salts was evaluated in BHI Broth adjusted to pH 7 before autoclaving, then a 0.3 percent *w*/*v* bile salt combination (Sigma-Aldrich, St. Louis, MO, USA) was added, and the medium was autoclaved. Saline solution (0.9 *w*/*v*) was used as a control.

A12, M4, and M10 isolates were then inoculated at 10^8^–10^8.5^ CFU/mL in each treatment and incubated at 28 °C at 50 rpm. Agar plate counts were performed for every hour for three hours on tryptic soy agar (TSA Scharlau, Spain), and plates were incubated at 28 °C for 36 h. The percentage of survival over time was estimated according to Equation (1).
(1)$$percentage\;of\;survival=\frac{bacterial\;concentration\;each\;treatment\;per\;hour\;\left(CFU/mL\right)}{bacterial\;concentration\;control\;at\;time\;0\;}\times 100$$

#### 2.4.5. Hydrophobicity Evaluation

The hydrophobicity of the isolates, as an indirect measure of adhesion ability, was determined using the Darilmaz, et al. [47] protocol. Briefly, two milliliters of the bacterial with an optical density of between 0.08 and 0.1 at 600 nm in saline solution (0.9 percent *w*/*v*) were vortexed for 1 min with 0.5 mL chloroform or 0.5 mL ethyl acetate (tests were performed independently with each solvent). They were then incubated for 10 min at 37 °C, the aqueous phase was removed, and the absorbance value change was measured. 

## 3. Results and Discussion

### 3.1. Whole-Genome Sequencing Identification

From the four assemblers (Megahit, Skesa, SPAdes, and Velvet) used, the ones made using SPAdes displayed better assembly characteristics (fewer contigs, the longest contig, the highest N50 index, and a small L50 index) (Appendix A). Then, using the ANIb, ANIm, and Tetra indices (calculated in JSpeciesWS), A12, M4, and M10 were identified as *L. lactis*, *Priestia megaterium*, and *Priestia* sp., respectively (Table 1). While phenotypic identification of probiotic bacteria using conventional microbiological techniques is critical for species identification, it is not always trustworthy. Furthermore, many species cannot be distinguished solely using these methods [22]. Lastly, reliable molecular techniques are required for species-level identification [23]. Numerous comparison assessments between two genome sequences, termed overall genome relatedness indices (OGRI), have been produced and proposed to serve as a cut-off or establish species boundaries [48]. Average nucleotide identity (ANI) is the most often utilized, with a proposed species boundary cut-off of 95–96% [49]. 

*L. lactis* was indeed the most abundant bacteria as detected by culture-independent methods (16S amplicon analyses) in the competitive exclusion culture we previously reported and used as a source for isolating these strains [25]. This result indicates that strain A12 also recovered as one of the important members of the competitive exclusion culture in the pure culture. Strains of this species have been used previously as a probiotic bacterium, mainly in mammals [50]. It exhibits different probiotic benefits such as enhancing resistance against various pathogenic bacteria (*Aeromonas hydrophila*, *Streptococcus iniae*), stimulation of the immune system (increasing respiratory burst activity, lysozyme activity, superoxide dismutase) and improving growth performance [51,52,53,54]. *L. lactis* strains isolated from Nile tilapia showed antibacterial activity against Gram-positive bacteria (*Lactobacillus plantarum*, *Staphylococcus aureus*, and *Enterococcus faecalis*), as well as against Gram-negative bacteria (*Vibrio* sp. and *Pseudomonas aeruginosa*) [55]. This inhibition could be related to the production of organic acids such as lactic, acetic, and butyric acids and inhibitory compounds such as class I bacteriocins of less than 55 kDa [56]. In addition, *L. lactis* has shown different benefits to the cultured fish, including improvement of growth performance, production of digestive enzymes, modulation of intestinal microbiota, resistance to diseases, and stimulation of the humoral immune response [57,58,59,60]. 

Analysis of strains M4 and M10, using genome-genome comparisons with state-of-the-art approaches and up-to-date genomic and taxonomic reference databases (TYGS), they were identified as *P. megaterium* for strain M4, while strain M10 clearly belongs to *Priestia* spp. genus but its genome and 16S rRNA gene sequence phylogeny shows that it represents a new previously undescribed species having as closest relatives both *P. megaterium* and *P. aryabhattai* (Appendix A). Therefore, we propose the name of Candidatus *Priestia unisabanae* sp. *nov.* for strain M10. Gupta, et al. [61] recently proposed the genus *Priestia* as a new classification for several type species of the genus *Bacillus* sp., including *Bacillus megaterium* and *Bacillus aryabhattai*. Nevertheless, since most of the published literature refers to them as *Bacillus megaterium* and *Bacillus aryabhattai*, this nomenclature will also be included here. *Bacillus* spp. used as probiotics exhibit antiviral activity, produce different exoenzymes (protease, amylase, and lipase), influence the intestine microbial populations, and enhance immune responses [62,63,64,65,66]. 

*B. megaterium* has been shown to remove toxic factors, improve liver function, and improve water quality parameters, microbial balance, immune response, disease resistance, growth, and enzyme production in fishery and aquaculture industries [55,58,65,67,68,69,70,71]. Some *Bacillus* strains are classified as human pathogens but *B. megaterium* has been considered safe according to the European Food Safety Authority because of its absence of enterotoxins and emetic toxins [68]. *B.*
*megaterium* is active against Gram-positive bacteria (*Listeria monocytogenes*, *Gardnerella vaginalis*, *Streptococcus agalactiae*, *Staphylococcus aureus*, and *Leuconostoc mesenteroides*), associated mainly with bacteriocin megacin production [68]. *B. aryabhattai* exhibits antibacterial action against *Vibrio* sp., the capability to modulate the gut microbiota, and immunostimulatory activity in *Litopenaeus vannamei* [72].

### 3.2. Pangenome Comparison

We reanalyzed the genomes and those downloaded from NCBI, which allowed us to determine that the assembled genomes of the isolated bacteria shared many of the same characteristics (genome length, number of coding sequences, number of rRNAs, and tRNAs) as the majority of genomes in the database (Appendix A). Using the Roary pangenome construction, a comparative genomic analysis was performed between the isolated microorganisms and other genomes available at the NCBI. As a result, it became clear that genomes of bacteria of the same species isolated from freshwater fish were not found in the analyzed database (Appendix A). Therefore, the genomes most closely related to A12, M4, and M10 were chosen using a phylogenetic tree of core genes (Figure 1a–c). The closest genomes were used to construct an upsetR plot, which revealed the presence of 254, 390, and 331 unique genes in A12, M4, and M10, respectively (Figure 2a–c). According to Melo-Bolívar, Ruiz Pardo, Hume and Villamil Díaz [19], probiotic microorganisms associated with the host gastrointestinal tract are more adapted to environmental changes in the host gastrointestinal tract, allowing these microorganisms to provide the expected beneficial effects. However, most probiotics used in aquaculture come from other sources (commercial or other animals), with only a small percentage isolated from the same cultured host.

### 3.3. Functional Annotation as an In-Silico Tool for Probiotic Screening in Aquaculture

There are different characteristics to selecting a probiotic candidate, including amino acid and vitamins biosynthesis, adherence capability, carbohydrates digestion, and bacteriocins production [55,73]. The present article determines the presence of genes related to amino acid and vitamins biosynthesis, adherent ability, carbohydrate utilization, and bacteriocin production. 

The functional annotation of the three bacteria’ genomes (A12, M4, and M10) revealed the existence of genes critical for their probiotic action, such as those involved in amino acid synthesis (glutamine, histidine, methionine, threonine, lysine, cysteine, leucine, glycine, and alanine) (Appendix A). Numerous studies have demonstrated that probiotics improve the host’s digestion by releasing digestive enzymes and growth-promoting components such as essential amino acids, fatty acids, and vitamins [66,74]. Since fish do not metabolize most essential amino acids, they must be added to the diet to improve growth rates [75].

Fish treated with probiotic supplemented diets may have a better Feed Conversion Ratio (FCR) because more gastrointestinal bacteria are working to break down nutrients, giving them more enzymes, vitamins, and amino acids [76]. According to Newsome, et al. [77], Tilapia may have been able to get the amino acids they needed from their gastrointestinal microbiota when they did not get enough from their food. The host and the microorganisms in the intestines further utilize amino acids [78]. 

Genes encoding fibronectin-binding proteins, elongation factor Tu, and chaperonin GroEL were detected in the genomes of three chosen bacteria. Additionally, *Lactococcus lactis* had genes encoding mucus and collagen-binding proteins (Appendix A).

Probiotic strains with adherent ability have a higher chance of colonizing the gastrointestinal tract [79]. In general, a large number of proteins released by probiotic microorganisms can bind to epithelial components such as mucus or ECM proteins (collagen, fibronectin, or laminin); these proteins include mucus-binding proteins, mannose lectin, mucus adhesion-promoting protein, elongation factor Tu, S-layer proteins, S-layer precursors, fibronectin-binding proteins, collagen-binding proteins, chaperon [80]. Additional proteins implicated in cell adhesion include those with the VWA (von Willebrand A) domain [81]. The genomes of *Lactococcus lactis* contained genes encoding VMA domain proteins involved in cell adhesion. Capsules, fimbriae, flagella, pilus, and fimbriae, are other structures that may participate in cell attachment [82]. Each of the three bacterial genomes contained genes encoding proteins involved in pilus production (Appendix A). Additionally, *P. megaterium* and *Priestia* sp. possessed genes involved in flagella development. All three bacteria were shown to have genes involved in cell attachment in the intestine (Appendix A). 

Also, Appendix A show the possible presence of genes related to the metabolism of carbohydrates in the three bacteria as sucrose, mannose, 2-ketogluconate, D-gluconate, fructose, D-ribose, xylose, L-arabinose, lactose, and galactose. 

The utilization of carbohydrates in the fish diet is critical to avoid protein or lipids from being catabolized for energy, resulting in retention and growth [83,84,85]. Furthermore, carbohydrates are the cheapest and most readily digestible primary energy source. Therefore, dietary carbohydrate supplementation for farmed aquatic animals can cover the animal’s nutritional needs while also avoiding lipids and protein as sources of energy [86]. In addition, digestive enzymes play a critical role in the hydrolysis of carbohydrates, lipids, and protein in fish diets and influence their feed efficiency [87]. According to previous research, increasing intestinal villi heights and widths may be due to the proliferation of probiotics and their attachment to the apical surface of epithelial cells, which improves carbohydrate utilization and short-chain fatty acid production in the digestive tract [88]. 

The genome of A12, M4, and M10 presented a gene related to bacteriocin production Lactococcin, Paeninodin, and Bacteriocin uviB, respectively Table 2. There are numerous mechanisms for eradicating fish pathogens, including production of bacteriocins, suppression of virulence gene expression of pathogenic bacteria, competition for adhesion sites, production of lytic enzymes, production of antibiotics, immunostimulation, competition for nutrients and energy, and production of organic acids [89].

### 3.4. Antibacterial Activity

As shown in Figure 3, the three potential probiotic bacteria evaluated, A12, M4, and M10, showed antibacterial activity against *S. agalactiae* and *A. hydrophila* compared to the control without treatment. As explained above, the probiotic with the highest inhibition of the two fish pathogens was A12, identified as *L. lactis*. Other studies show that *L. lactis* isolated from Nile tilapia have antibacterial activity against Gram-positive bacteria (*L. plantarum*, *Staphylococcus aureus*, and *Enterococcus faecalis*) and gram-negative bacteria (*Vibrio* sp. and *Pseudomonas aeruginosa*) [90,91]. The inhibition could be related to the production of organic acids, such as lactic, acetic, and butyric acid, and the production of a bacteriocin of less than 5 kDa with class I bacteriocins detected [56]. On the other hand, *B. megaterium* is more active against gram-positive bacteria including human pathogens (*Listeria monocytogenes*, *Gardnerella vaginalis*, *S. agalactiae*, *Staphylococcus aureus*, and *Leuconostoc mesenteroides*), associated mainly with bacteriocin megacin production [68]. 

### 3.5. Antibiotic Minimal Inhibitory Concentration

Antibiotics Streptomycin, Ciprofloxacin, Kanamycin, Tetracycline, Ampicillin, Vancomycin, Clindamycin, Gentamicin, and Chloramphenicol were tested on *L. lactis* A12, *P. megaterium* M4, and *Priestia* sp. M10 to identify their minimal inhibitory concentrations by Etest strips (Biomerieux, Craponne, France). From this, it was determined that *L. lactis* A12 is resistant to Tetracycline, and *Priestia* strains M4 and M10 are resistant to Clindamycin (Table 3). Antibiotic resistance and a growing reluctance to administer antibiotics have led to an increase in the use of probiotics [4]. Bacterial antibiotic resistance mechanisms can be innate, natural, or acquired. There is no horizontal transferability of the intrinsic resistance; nevertheless, the acquired resistance can be gained by mutations or the acquisition of genes through mobile genetic elements into their genomes [92,93].

Furthermore, the altered DNA can be transferred via different genera and species by conjugative plasmids and mobile elements, which spread the resistance in microbial communities [94]. In other words, candidate probiotics should not contain transmissible antibiotic-resistant genes, which can increase the risk of disease agents developing antibiotic resistance [95]. For example, a research-tested 473 LAB isolates from the genera *Lactobacillus*, *Pediococcus*, and *Lactococcus* showed that 17 *Lactobacillus* isolates were resistant to at least one tested antibiotic [96]. More commonly, genes giving resistance to tetracycline, erythromycin, and vancomycin are discovered in lactococci, enterococci, and lactobacilli isolated from fermented meat and milk products, likely because of substandard veterinary methods [97,98]. For example, it was discovered that *Lactococcus* (*L. lactis*) strains were resistant to tetracycline and erythromycin [99].

As a side note, antibiotic-resistant probiotics may help restore beneficial microbes in the intestine following antibiotic treatments, but such resistance should not be delivered via plasmid [68,95]. Also, according to Jose, et al. [100], antibiotic resistance by probiotic strains poses no risk to food safety or consumer health as long as resistance genes are non transmissible. 

Some alternatives to continue with the study of probiotics with antibiotic resistance include their use as paraprobiotics or a technique to eliminate the antibiotic genes contained in a plasmid from the probiotic bacteria. On one side, together with recent findings that non-viable bacteria are beneficial to hosts like their viable complement, these factors have prompted the search for non-viable probiotic formulations, often known as paraprobiotics. Due to their capacity to remain stable throughout a wide range of temperatures and pH, paraprobiotics may be of significant relevance in the dairy industry, allowing their inclusion in meals with maximum acidity and before thermal processing without functionality loss [98]. On the other side, antibiotic resistance could be eliminated from probiotic strains by utilizing techniques that did not genetically modify the organism and did not alter the probiotic properties [97,98]. For example, Rosander, et al. [101] have shown that *Lactobacillus reuteri* ATCC 55730 may be cleansed of two separate plasmids harboring undesirable antibiotic resistance features by the protoplast formation curing method, and the resulting *L. reuteri* DSM 17938 strain retains all its probiotic properties.

### 3.6. Hemolytic Activity

As illustrated in Figure 4, the three bacteria were γ hemolysis. In other words, they are non-hemolytic microorganisms because the red blood cells surrounding the colonies obtained during growth were not lysed. Numerous microorganisms can produce exotoxins that can induce partial (α hemolysis, such as *Streptococcus pneumoniae*) or complete (β hemolysis, such as *Streptococcus dysgalactiae*) lysis of human or animal erythrocytes. β hemolysis is shown by a clear zone around the colonies, α hemolysis is shown by a greenish zone around the colonies, and γ hemolysis is shown by no clear zone around the colonies, which is a safe non-hemolytic isolate [44,102].

### 3.7. Bile Salts and pH Survival

*P. megaterium* M4 and *Priestia* sp. *M10* demonstrated the highest survival levels in bile salts and low pH. *M10* showed the highest survival levels for bile salts and low pH during the three-hour evaluation period (Figure 5b,c,e,f,h,i). Furthermore, *L. lactis* A12 presented a 64%, a 67%, and a 41% reduction in survival after being exposed to pH 2, pH 3, and bile salts, respectively, during the first hour following exposure to each of the three conditions (Figure 5a,d,g). M4 and M10 showed the highest gastrointestinal survival based on the test evaluated compared to A12. Probiotics have aided in the improvement of water quality, the increase of dietary nutrient utilization through the development of supplemental digestive enzymes, the reduction of disease incidence, the rise of survival, the improvement of the immune response, and the modulation of microbial colonization in aquaculture [83,103]. The source of the bacteria, their safety, and the strains’ ability to survive through the host’s gastrointestinal tract (e.g., resistance to bile salts and low pH) are all critical factors in determining whether a microorganism should be classified as probiotic [57,58,104,105,106]. Alonso, et al. [107] show that strains of lactococci are more sensitive to concentrations between 0.5 and 1 percent of bile salts, growing exclusively in broth supplemented with 0.1% bile salts. Other microorganisms like *Lactobacillus* (*Lactobacillus gasseri* TSU3 and *Lactobacillus animalis* TSU4) demonstrated substantial tolerance (*p* < 0.001) to bile salt after 24 h, with findings equivalent to the probiotic strain of reference, *Lactobacillus acidophilus* NCDC15 [108]. In contrast, a *L. lactis* strain isolated from wild fish has more tolerant of high acidic (like pH 2) and bile salt concentrations than counterparts from terrestrial sources [109]. Similarly, 5 h of incubation at a pH of 2.5 did not affect the survivability of *Lactobacillus gasseri* TSU3 and *Lactobacillus animalis* TSU4. Viability was 10^9^ CFU ml^−1^, the same as the initial time point [108]. 

### 3.8. Hydrophobicity Evaluation

Table 4 summarizes the hydrophobicity percentages of the three strains studied. It was found that the results vary according to the organic solvent employed and the species of bacterium; *L. lactis* A12 exhibited higher hydrophobicity when chloroform was used than when ethyl acetate was used, at 92% and 79%, respectively. Similarly, *Priestia* sp. M10 exhibits higher hydrophobicity to chloroform (83%) but a relatively lower hydrophobicity to ethyl acetate (23%). M4 has the lowest hydrophobicity to chloroform and ethyl acetate, with 29% and 0%, respectively. In summary, *L. lactis* A12 and *Priestia* sp. M10 exhibited greater hydrophobicity than *P. megaterium* M4, indicating a more stable potential capacity to cling to the host intestinal surface. 

The ability of probiotic strains to adhere and colonize the gut cells is crucial to exclude potentially harmful microorganisms and exert the desired activity [110,111,112]. The hydrophobicity of bacteria is a measure of the capacity of probiotics to adhere to the intestinal mucosa [111]. According to Ghori, et al. [113], the greater the hydrophobicity, the stronger the adhesion to the intestinal epithelium. The hydrophobicity of the cell surface is a non-specific relationship between microbial cells and their hosts. The first contact may be weak and reversible and occurs before the emergence of more complicated mechanisms involving cell surface proteins and lipoteichoic acids that mediate subsequent adhesion processes [114]. 

Meidong, Doolgindachbaporn, Jamjan, Sakai, Tashiro, Okugawa and Tongpim [111] showed that *Bacillus siamensis* strain B44v exhibited a higher cell surface hydrophobicity (64.8 percent) compared to *Bacillus* sp. strain B51f (42.9 percent). *B. subtilis* HAINUP40 showed considerable hydrophobicity on its cell surface in xylene (28.8 percent) and chloroform (41.3 percent) [105]. In the presence of xylene, the hydrophobicity of the cell surface of *Enterococcus hirae* F2 was 38.7 percent, which was greater than that of n-hexadecane, which was 34.4 percent [115]. *B. subtilis* E221 exhibited greater hydrophobicity in the presence of xylene (37.22%) and chloroform (45.67%) than in the presence of ethyl acetate (17.92 percent) [116]. *Bacillus amyloliquefaciens* FC6 had the highest adhesion capacity (86.65 percent in xylene, 69.14 percent in ethyl acetate, 65.71 percent in chloroform, and 57.07 percent in toluene), followed by *B. subtilis* FS1 (45.08 percent in chloroform, 39.91 percent in ethyl acetate, 29.79 percent in toluene) and *Bacillus cereus* FC3 (43.1% in toluene, 41.66% in xylene, 35.86% in chloroform and 28.89% in ethyl acetate) [117].

In summary, *L. lactis* A12 and *Priestia* sp. M10 have the highest adhesion to the intestinal epithelium, as said by Liu, Wang, Cai, Guo, Cao, Zhang, Liu, Yuan, Zhu, Zheng, Xie, Guo and Zhou [105] hydrophobicity properties determine the interactions among probiotic bacteria and host epithelial cells. In addition, the hydrophobic surface is responsible for bacterial attachment to host cells [115].

Our findings seem to suggest that we isolated microorganisms with probiotic properties using the competitive exclusion culture derived from the tilapia gut microbiome. As a proof of concept, we characterized three promising bacterial isolates, identified as *Lactococcus lactis* A12, *Priestia megaterium* M4, and *Priestia* sp. M10 (new undescribed species), expanding the set of strains of the digestive tract of Tilapia that can serve as autochthonous probiotics of Tilapia from their symbiotic bacteria, with the multiple advantages in adaptation and engraftment this approach may have. After comparing the pangenomes with the related genomes from NCBI, it shows the presence of unique genes as well as phenotype of their potential survival in the gastrointestinal of fish that may play an important role in the adaptation to the fish environment and the antagonistic activities/antibiosis they have against common bacterial aquaculture pathogens. Also, the genome annotation showed the presence of essential genes related to the biosynthesis of amino acids, vitamins, bacteriocin-like proteins, adhesion genes, and metabolism of carbohydrates, summing up the potential probiotics application of these three strains. In vitro evaluation of the probiotic potential of these strains showed favorable features for their use as probiotics in tilapia culture. However, in vivo experimentation at different production scales are among the most important next steps, as well as the presence of antibiotic resistance genes detected in the isolated microorganism requires further evaluation of how contained these resistances are and whether they can experience horizontal transfer, and assessment if there is any preferential suitability of paraprobiotic vs. probiotic preparations in this sense.

We are providing a proof of concept regarding the use CFCEC originating from the Tilapia gut microbiome as a way to raise and recover symbiotic bacteria from a host aimed to be protected, proving that probiotic strains representing key members of the consortia exhibiting competitive exclusion features can be recovered in pure cultures, they represent novel bacterial diversity. CFCEC originating from Tilapia gut microbiome can be used to screen symbiotic bacteria which then further genotypic and phenotypic and functional analysis can define them as potential probiotics. Field experimentation is higly recommended to further proof the biological, physiological and economical benefits of these autochthonous bacteria strains in Tilapia fisheries. Furthermore, some of these autochthonous strains exhibit desirable activities that may protect for more healthy and productive growth when used as probiotics the deliberately targeted source and host (Tilapia) in a more resilient, adapted, and customized manner than the usual probiotic application of allochthonous strains where the effectivity has a greater incidental/fortuitous factor. 

## 4. Conclusions

The primary conclusion of this study is that a continuous flow competitive exclusion culture developed from the microbiota of Nile tilapia is a rich source of autochthonous probiotic bacteria. Moreover, these bacteria contain a number of unique genes that could improve the health and nutrition of the host. Thus, they can contribute to the Nile tilapia farming industry. In order to continue the development of the probiotic product, the next step is to design the probiotic consortium, incorporate these bacteria into fish feed, and assess the effect on growth performance, microbiota modulation, intestinal morphology, and immunomodulation.

## Figures and Tables

**Figure 1 microorganisms-10-01376-f001:**
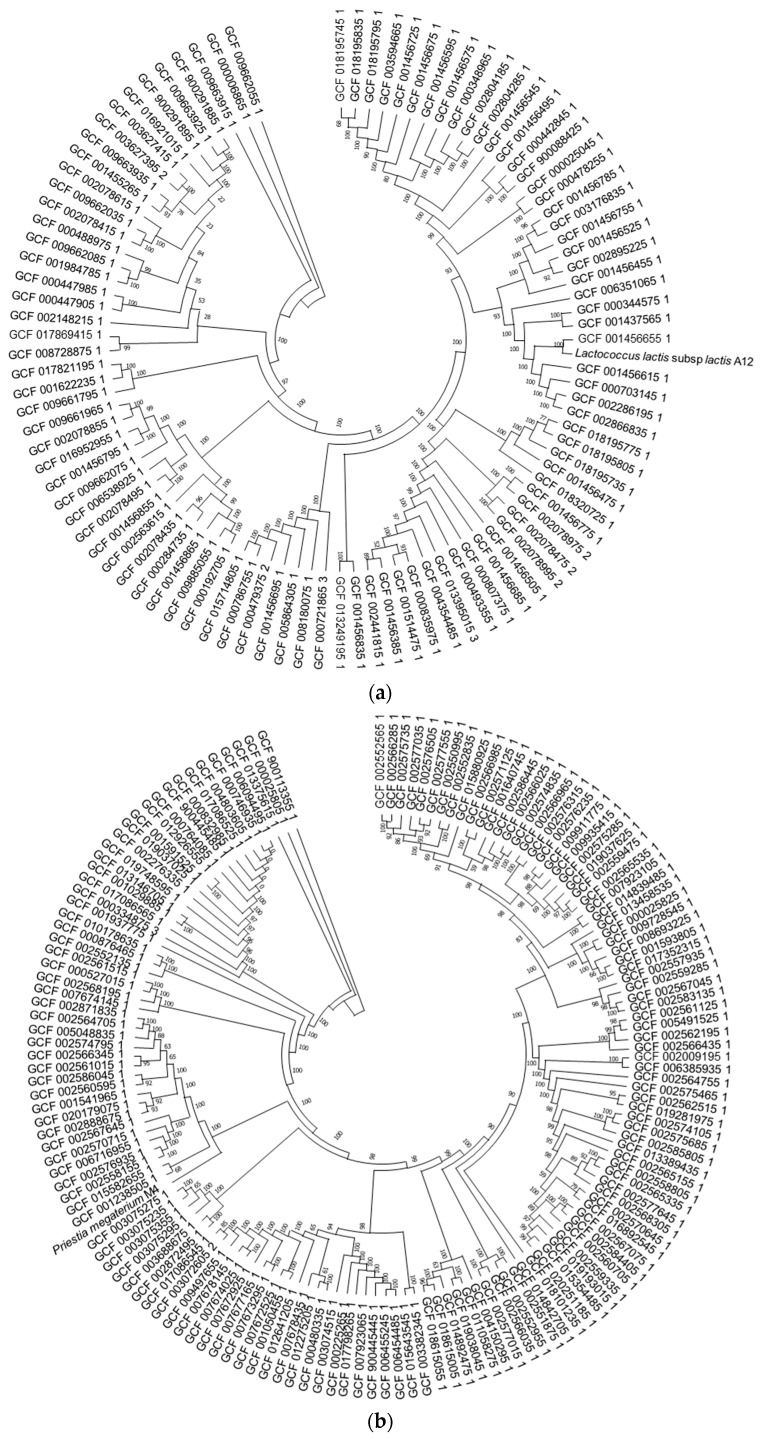
Core-genome maximum-likelihood phylogeny of *Lactococcus lactis* subsp. *lactis* (**a**), *Priestia megaterium* (**b**), and *Priestia aryabhattai* (**c**) strain from NCBI (RefSeq Database, https://www.ncbi.nlm.nih.gov/refseq/ (accessed on 9 January 2022)) and *Lactococcus lactis* A12, *Priestia megaterium* M4, and *Priestia* sp. M10, respectively. A core-genome phylogenetic representation using IQ-Tree. The tree was built from core genes identified by Roary per each strain. The log-likelihood score for the consensus tree was constructed from 1000 bootstrap.

**Figure 2 microorganisms-10-01376-f002:**
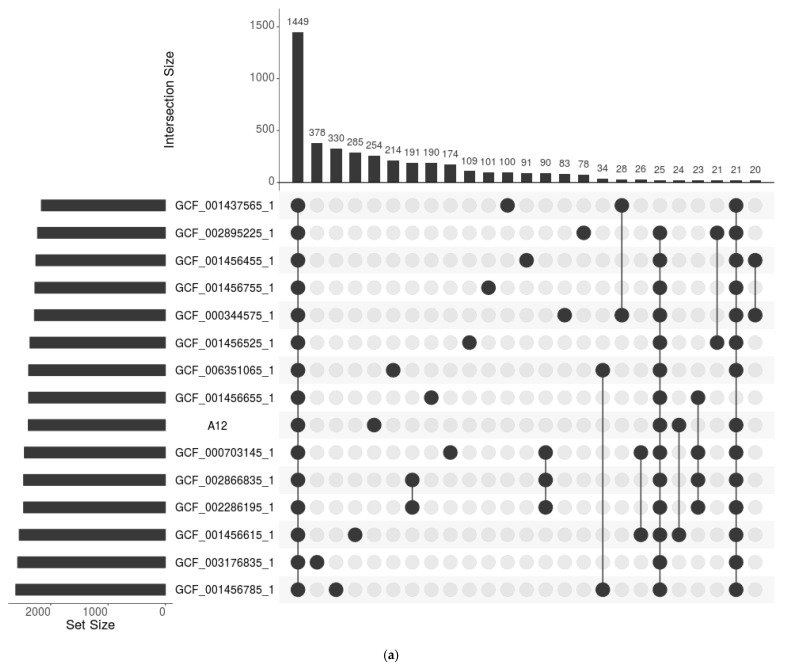
UpSetR plot of Roary genes presence/absence file from the closest related genome to *Lactococcus lactis* A12 (**a**), *Priestia megaterium* M4 (**b**), and *Priestia* sp. M10 (**c**).

**Figure 3 microorganisms-10-01376-f003:**
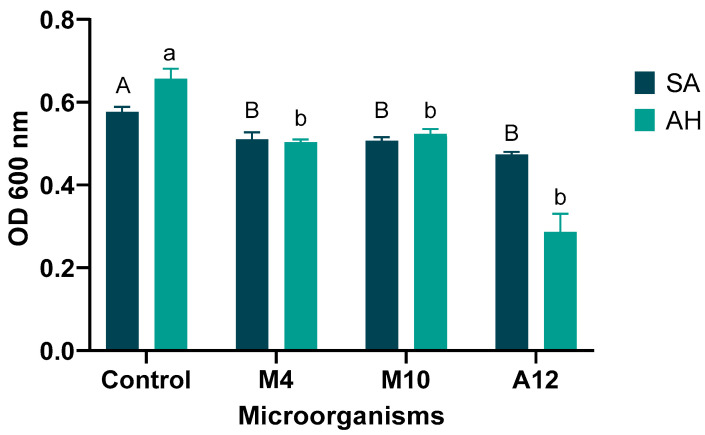
Antibacterial activity of the cell-free extracellular products (CFECP) of *L. lactis* A12, *P. megaterium* M4, and *Priestia* sp. M10 against *S. agalactiae* (SA) and *A. hydrophila* (AH). Capital letters indicate significant differences between each probiotic against the control in *Streptococcus agalactiae* treatment (*p* < 0.05). Lower-case letters indicate significant differences of each probiotic against the control in *Aeromonas hydrophila* treatment (*p* < 0.05). Data represent mean ± SEM (*n* = 3).

**Figure 4 microorganisms-10-01376-f004:**
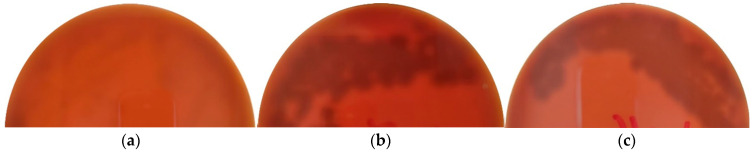
Hemolysis analysis picture results for (**a**) *L. lactis* A12, (**b**) *P. megaterium* M4, (**c**) *Priestia* sp. M10.

**Figure 5 microorganisms-10-01376-f005:**
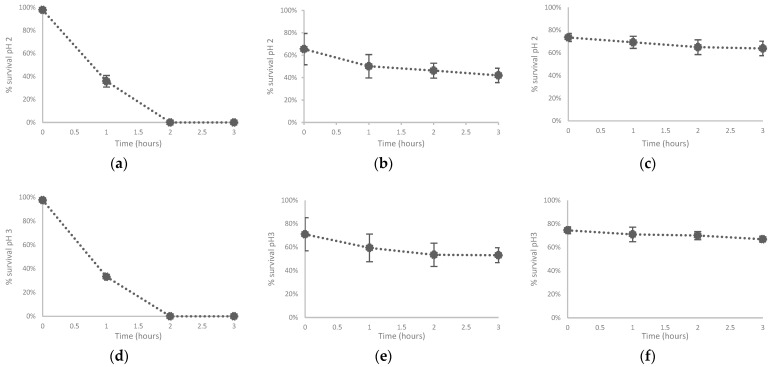
Evaluation of the survival to gastrointestinal conditions such as low pH (2 and 3) and bile salts of *L. lactis* A12, *P. megaterium*, *Priestia* sp. M10. (**a**): % survival pH 2 *L. lactis* A12, (**b**): % survival pH 2 *P. megaterium* M4, (**c**): % survival pH 2 *Priestia* sp. M10, (**d**): % survival pH 3 *L. lactis* A12, (**e**): % survival pH 3 *P. megaterium* M4, (**f**): % survival pH 3 *Priestia* sp. M10, (**g**): % survival bile salts *L. lactis* A12, (**h**): % survival bile salts *P. megaterium* M4, (**i**): % survival bile salts *Priestia* sp. M10.

**Table 1 microorganisms-10-01376-t001:** Determination of the overall genome relatedness indices (ANIm, ANIb, Tetra) of the bacteria A12, M4, and M10.

Bacteria	ANIb	ANIm	Tetra
A12	*Lactococcus lactis* subsp. *lactis* ATCC 19435 [T] (97.1%)	*Lactococcus lactis* subsp. *lactis* JCM 5805 = NBRC 100933 [T] (98.1%)	*Lactococcus lactis* subsp. *lactis* ATCC 19435 [T] (0.998)
M4	*Priestia megaterium* Riq5 (96.46%)	*Priestia megaterium* Riq5 (97.39%)	*Priestia aryabhattai* LK25 (0.999)
M10	*Priestia aryabhattai* LK25 (98.96%)	*Priestia aryabhattai* LK25 (99.31%)	*Priestia aryabhattai* LK25 (0.999)

**Table 2 microorganisms-10-01376-t002:** Gen related to Bacteriocin-like protein in *Lactococcus lactis* (A12), *Priestia megaterium* (M4), and *Priestia* sp. (M10).

Bacteria	Gen	NCBI Reference	Scaffold	Position	Size (nt)	Size (aa)	Strand
A12	Lactococcin family bacteriocin	WP_153242284.1	1	2090503-2090715	213	71	+
M4	Paeninodin family lasso peptide	WP_014461234.1	1	1055241-1055366	126	42	+
M10	Bacteriocin uviB	WP_097813194.1	29	13487-13263	225	75	−

**Table 3 microorganisms-10-01376-t003:** Minimal Inhibitory Concentration probiotic *Lactococcus lactis* A12 (A12), *Priestia megaterium* M4 (M4), *Priestia* sp. M10 (M10).

	A12	Cut-Off Value *	M4	Cut-Off Value **	M10	Cut-Off Value **
Streptomycin	24–32 (S)	32	0.25 (S)	8	0.5–0.75 (S)	8
Ciprofloxacin	0.75	not reported	0.47	not reported	0.64	not reported
Kanamycin	8 (S)	64	0.94 (S)	8	0.125 (S)	8
Tetracycline	≥256 (R)	4	0.25 (S)	8	0.5 (S)	8
Ampicilin	0.094–0.125 (S)	2	0.25–0.50	n.r	0.125–0.19	n.r
Vancomycin	0.50 (S)	4	0.19 (S)	4	0.75–1.5 (S)	4
Clindamycin	0.25–0.38 (S)	1	32 (R)	4	48 (R)	4
Gentamicin	1.5–2.0 (S)	32	0.016 (S)	4	0.023 (S)	4
Cloramphenicol	3.00 (S)	8	3 (S)	8	2 (S)	8

* Cut off values by EFSA [42] (µg/mL) *Lactococcus lactis* strain. ** Cut off values by EFSA [42] (µg/mL) *Bacillus* spp. strain. Resistant (R) MIC > cut-off value. Susceptible (S) MIC ≤ cut-off value.

**Table 4 microorganisms-10-01376-t004:** Hydrophobicity evaluation of A12, M4, and M10.

Bacteria	Chloroform (%)	Ethyl Acetate (%)
*L. lactis* A12	92 ± 0.02	79 ± 0.01
*Priestia* sp. M10	83 ± 0.04	23 ± 0.19
*P. megaterium* M4	29 ± 0.07	0 ± 0.00

## Data Availability

The raw reads used to assemble the draft genome were deposited in the Sequence Read Archive (SRA) under Bioproject accession number PRJNA577063. The genome sequence data of M10, A12 and M4 are deposited with Accession numbers of SRX15182551, SRX6979445, SRX6979446, respectively.

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
