# Peer review of "Competitive Exclusion Bacterial Culture Derived from the Gut Microbiome of Nile Tilapia (Oreochromis niloticus) as a Resource to Efficiently Recover Probiotic Strains: Taxonomic, Genomic, and Functional Proof of Concept"

_microorganisms, 2022, doi:10.3390/microorganisms10071376_

Round 1

Reviewer 1 Report

The manuscript written on recovering probiotic strains from the gut of Nile tilapia as CFCEC is interesting. However, I found several shortcomings in the manuscript.

The outcome of the study given in abstract is very general such as  

These results were consistent with reports on strains inside these phyla exhibiting probiotic features, and the strains we found are expanding their known diversity. “

Authors need to give specifics, how they found that it is expanding diversity, what evidence they have?  How they say it is consistent?

Also, there is no conclusion drawn in the abstract.

Overall, I am not convinced with the way abstract is presented..

You can include this as one of the keywords:  continuous-flow competitive exclusion 14 consortia (CFCEC)

The purpose of the study is introduced well in Introduction section.  The explanation on the expansion of work is good.

Regarding ethical, permission was obtained on 26 May 2015, how long this is valid?

In results, The statements given from “The Shovill pipeline (https://github.com/tseemann/shovill) was used….species-level identification [23]” are more appropriate to be given in methods section. Result section should be utilized for the outcome of research. In your case, even if it is results and discussion together, it would be better to discuss the outcome of this research with scientific literature rather than the methodology used.

There is abundant literature apart from reference 25 to write and analyze about L. lactis, use them.

Table S2, feature ID links are not working. Also, since the table is very big and to make user friendly, I suggest giving expansion of the abbreviation somewhere in the table heading instead of caption. Also, it is more ideal to expand them especially those that used in ‘type’ section.

Figure 3, legend, capital letters indicate significant difference, … which capital letter, B or A, if B represent probiotic against respective control, what does the A refers. What comparison you did for A and a?

The conclusion section is too big. I suggest, it would be better, if authors desire, they can introduce a summary section and give all the outcomes with justification and finally, in conclusion, they can give a take home message in brief.

To summarize, I enjoyed reading this manuscript.

Author Response

Reviewer #1 (comments for the author (Required))

  1. The outcome of the study given in abstract is very general such as “These results were consistent with reports on strains inside these phyla exhibiting probiotic features, and the strains we found are expanding their known diversity. “ Authors need to give specifics, how they found that it is expanding diversity, what evidence they have?  How they say it is consistent?

R/ We are referring to the results described in detail in the main text about comparative genomics: in this case, while we can assign a rather precise taxonomic affiliation to the strains, it is clear they contain and represent new variants inside those genera, and there is even a well-supported case of a representative of a new species in the isolates obtained (with differences in gene content and putative functions inferred)

  1. Also, there is no conclusion drawn in the abstract.

R/ According to reviewer 1, the conclusion was included as follows: "In conclusion, CFCEC proved to effectively allow the enrichment and further pure culture isolation of strains with probiotic potential." (lines 27-28)

  1. Overall, I am not convinced with the way abstract is presented..

R/ We understand the opinion and we carefully thought about ways on how to find a common balance between the reviewers comments. We worked hard and we think that the abstract reflects the rationale and findings presented in the manuscript.

  1. You can include this as one of the keywords:  continuous-flow competitive exclusion 14 consortia (CFCEC)

R/ Thanks, we follow the suggestion of reviewer 1, and the keywords continuous-flow competitive exclusion culture (CFCEC) were included in line 29.

  1. Regarding ethical, permission was obtained on 26 May 2015, how long this is valid?

R/ Date of the permit was corrected and the time authorised was included as follows: "The project followed the Colombian national government's regulations. The Permit for accessing genetic resources was issued by the Colombian Ministry of Environment Number 117 (Otrosí 4) on the 8th of May 2018 for five years." (lines 93-95)

  1. In results, The statements given from “The Shovill pipeline (https://github.com/tseemann/shovill) was used….species-level identification [23]” are more appropriate to be given in methods section. Result section should be utilized for the outcome of research. In your case, even if it is results and discussion together, it would be better to discuss the outcome of this research with scientific literature rather than the methodology used.

R/ The sentences including methodology were removed as follows: "From the four assemblers (Megahit, Skesa, SPAdes, and Velvet) used, the ones made using SPAdes displayed better assembly characteristics (fewer contigs, the longest contig, the highest N50 index, and a small L50 index) (Table S1)." (Lines 239-240)

  1. There is abundant literature apart from reference 25 to write and analyze about L. lactis, use them.

R/ We are agree with reviewer 1 about the different literature published about Lactococcus lactis, however, we used reference 25 to indicate the source of the bacteria isolated in the present study.

  1. Table S2, feature ID links are not working. Also, since the table is very big and to make user friendly, I suggest giving expansion of the abbreviation somewhere in the table heading instead of caption. Also, it is more ideal to expand them especially those that used in ‘type’ section.

R/ According to reviewer 1, in Table S2, Table S3, and Table S4, the feature Ids was removed because the links are not working without access to a RAST account. Also, the type column was removed because we considered that it is repeated information and all of the genes are CDS.

  1. Figure 3, legend, capital letters indicate significant difference, … which capital letter, B or A, if B represent probiotic against respective control, what does the A refers. What comparison you did for A and a?

R/ Capital letters represent the significant statistical difference of each probiotic against the control letter A is the control group and letter B indicate that the three probiotics have significant statistical support against the control. We do not compare capital letter "A" with lowercase letter "a" because they are different treatments. Capital letter are Streptococcus agalactiae and lowercase are Aeromonas hydrophila

  1. The conclusion section is too big. I suggest, it would be better, if authors desire, they can introduce a summary section and give all the outcomes with justification and finally, in conclusion, they can give a take home message in brief.

R/ Following the comment of reviewer 1, a summary section in the last part of the research and discussion section was included as follow "Our findings seem to suggest that we isolated microorganisms with probiotic…   … and customized manner than the usual probiotic application of allochthonous strains where the effectivity has a greater incidental/fortuitous factor." Also, a short conclusion paragraph was included to help the readers to identify the main message: "The primary conclusion of this study is that a continuous flow competitive exclusion culture developed from the microbiota of Nile tilapia is a rich source of autochthonous probiotic bacteria. Moreover, these bacteria contain a number of unique genes that could improve the health and nutrition of the host, and could contribute to the Nile tilapia farming industry. In order to continue the development of the probiotic product, the next step is to design the probiotic consortium, incorporate these bacteria into fish feed, and assess the effect on growth performance, microbiota modulation, intestinal morphology, and immunomodulation.".  (Lines 535 -611)

Reviewer 2 Report

The authors refer to FAO 2017 and 2019 data in the introduction. Data for 2020 should be added.

The article’s text contains references to electronic information resources. These references should be placed in the References section.

In general, the article has a high applied significance of the research and corresponds to current scientific trends.

Author Response

Reviewer #2 (comments for the author (Required))

  1. The authors refer to FAO 2017 and 2019 data in the introduction. Data for 2020 should be added.

R/ Following reviewer 2 remark, FAO citation is updated as follows: "More than 61.04 million people worldwide are directly involved in the fishing industry, with fishing accounting for 63.4% and aquaculture accounting for 36.6% [1]. " (Lines 35-37)

  1. The article’s text contains references to electronic information resources. These references should be placed in the References section.

R/ We are now including in the revised version the web page references requested as follows: Shovill v1.1.0 (SPAdes, v3.15.3; Velvet, v1.2.10; Megahit, v1.2.9; Skesa,v2.4.0; using default arguments; https://github.com/tseemann/shovill) [26] was used to perform quality control (QC), filtering, trimming, and de novo assembly on raw WGS data (FASTQ files). (Lines 128-130)

Then, the assembled contigs were uploaded with their respective reference genomes to Medusa (http://combo.dbe.unifi.it/medusa) [26] to determine orientation and order among contigs to produce longer scaffolds [29,30]. (Lines 142-145)

Information on nomenclature, synonymy, and associated taxonomic literature was provided by TYGS's sister database, the List of Prokaryotic names with Standing in Nomenclature (LPSN, available at https://lpsn.dsmz.de) [31]. (Lines 148-151)

To find genes involved in amino acid biosynthesis, vitamin biosynthesis, carbohydrate metabolism, adhesion, and aggregation, among other features,  Uniprot database (https://www.uniprot.org/) [35] (Lines 167-169)

A Blastx (https://blast.ncbi.nlm.nih.gov/Blast.cgi) [39] was used to reannotate the gene identified in our genomes (Lines 172-173)

Reviewer 3 Report

Dear authors,

The manuscript Competitive exclusion bacterial consortia derived from the gut microbiome of Nile tilapia (Oreochromis niloticus) as a resource to efficiently recover probiotic strains: taxonomic, genomic, and functional proof of concept presents the results of the genomic identification of three strains recovered from the tilapia microbiome. They also describe some tests carried out on these strains to verify their possible use as probiotic microorganisms in fish. However, the methodology does not mention the controls used, and the discussion and conclusions should be improved. There are several details that the authors should clarify and improve in the manuscript.

1.       Which is the objective of the manuscript? The objective is not clear.

2.       On the methodology:

a.                  Which antibiotics were used for the minimal inhibitory concentration test?

b.                  What does mean ECPs?

c.                   Which strains were used as control in hemolysis test and pH and bile salts survival?

d.                  References Aragon-Rosas and Muthukumar and Kandeepan are missing

e.                  References cited in Bile salts and pH survival (41 and 42) are wrong.

3.       In the results and discussion section, the discussion of the results should be improved, not their description or explanation, but the discussion itself (in comparison with other publications).

4.       It is not clear to me what the final conclusion of the manuscript is.

I have comments and corrections marked in the text for you. Please make them.

Kind regards

Author Response

Reviewer #3 (comments for the author (Required))

  1. Which is the objective of the manuscript? The objective is not clear.

R/ The objective, in line with sentences on lines xx-xxx, is to show how the previously developed strategy CFCEC to recover potential probiotic strains using Tilapia microbiome as source is actually an efficient selective isolation way to obtain strains with probiotic beneficial features: the strains are now studied at the genomic level and activity detail, providing further evidence of the advantages of this approach to recover autochthonous isolates that are already adapted to the targeted host for improved probiosis. We also remark how this is a characterization step for this set of strains that pave the way for further stages testing improved health and growth in Tilapia fish cohorts under mixed inocula, and this would be promising research paths to explore.

  1. aerobic or microaerobic atmosphere?

R/ According to reviewer 3 suggestion, we modified the sentence to detail the atmosphere used: "These bacteria were activated by inoculating them onto Tryptose Soy Agar (TSA, Scharlab S. L., Barcelona, Spain) and incubated at aerobic atmosphere for 24 h at 28°C in preparation for further use." (Line 104)

  1. Antibacterial activity against Streptococcus agalactiae and Aeromonas hydrophila. Italics

R/ Bacterial names are corrected in format (italics) as follows:  "Antibacterial activity against Streptococcus agalactiae and Aeromonas hydrophila" (Line 180)

  1. Which antibiotics were used for the minimal inhibitory concentration test?

R/ Answering the request of reviewer 3, details on antibiotics used for the minimal inhibitory concentration test were included as follows: "The minimal inhibitory concentration of bacteria against selection of antibiotics (Streptomycin, Ciprofloxacin, Kanamycin, Tetracycline, Ampicilin, Vancomycin, Clindamycin, Gentamicin, Cloramphenicol) was evaluated according to Florez, et al. [41]. ". (Lines 192-194)

  1. What does mean ECPs?

R/ Complete name of ECPs was included as follows: "Then, the bacterial culture was adjusted to pH 3.5 with 2N HCl. After that, the culture was heated at 80°C and centrifuged at 500 x g for 30 min. Followed by adjusting the extracellular products (ECPs) to pH 6.5 and the 0.22 µm filter was used to sterilize" (Lines 183-186)

  1. Which strains were used as control in hemolysis test and pH and bile salts survival?

R/ We are agree with the reviewer 3 about the important of using some strains as a control, however, in the present results we did not include that control, and we relied on peer-reviewed published results when we mentioned other probiotic strains results with the corresponding citation.

  1. References Aragon-Rojas and Muthukumar and Kandeepan are missing.

R/ Bibliographic references are now corrected as follows: "With some changes, this test was conducted following Muthukumar and Kandeepan [41], and Aragón-Rojas, et al. [42] Muthukumar and Kandeepan [41], and Aragón-Rojas, et al. [42]. " (Lines 217-218)

  1. How can you do a quantitative survival test if you are using a highly enriched medium, what makes you assume that the bacteria did not multiply despite the pH or the concentration of bile salts used?Which strains did you use as control?

R/ We agree with reviewer 3 comments about how enriched medium could cause the growth of the bacteria, however, as shown in Figure 5, there is no growth of the microorganisms, but a reduction of the colony forming units. Therefore, the survival equation was used to estimate the number of bacteria surviving after the treatment with a pH or bile salts. The enriched medium test allowed us to differentiate that the effect over the bacteria is only from the pH changes or bile salts concentration. We only used saline solution as control in order to ensure that there was no bacteria growth from new carbon sources. We did not use other bacterial strains as positive or negative control, but the internal controls of the assays with the same experimental subjects, and we restrict our analyses and conclusions to this precise setup.

  1. Lactobacillus, italics

R/ Bacterial names were corrected in formatting (italics) as follows: "For example, a research-tested 473 LAB isolates from the genera Lactobacillus, Pediococcus, and Lactococcus showed that 17 Lactobacillus isolates were resistant to at least one tested antibiotic [96]. " (Lines 423-425)

  1. Numerous microorganisms, such as?

R/ According with reviewer 3 suggestion, example of each type of hemolysis was included as follows: "Numerous microorganisms can produce exotoxins that can induce partial (α hemolysis, such as Streptococcus pneumoniae ) or complete (β hemolysis ,such as Streptococcus dysgalactiae) lysis of human or animal erythrocytes." (Lines 458-460)

  1. In the results and discussion section, the discussion of the results should be improved, not their description or explanation, but the discussion itself (in comparison with other publications).

R/ Comparison with other studies recovering strains with probiotic features are mentioned and cited in the introduction, and the differential approach we used to recover them is quite particular, thus, we focus our discussion mainly restricted to the findings and analyses, but also there are some sentences about the foreseen plans now that the strains are proving to be indeed promising for the application, where many publications are reported, and we will have then the opportunity to actually compare the effects of inocula in the growth and health parameters of Tilapia fish cohorts to similar studies.